# Diagnostic Performance of Height-Estimated Baseline Creatinine in Diagnosing Acute Kidney Injury in Children with Type 1 Diabetes Mellitus Onset

**DOI:** 10.3390/children9060899

**Published:** 2022-06-16

**Authors:** Stefano Guarino, Giulio Rivetti, Anna Di Sessa, Maeva De Lucia, Pier Luigi Palma, Emanuele Miraglia del Giudice, Cesare Polito, Pierluigi Marzuillo

**Affiliations:** Department of Woman, Child and of General and Specialized Surgery, Università degli Studi della Campania “Luigi Vanvitelli”, Via Luigi De Crecchio 2, 80138 Naples, Italy; stefano.guarino@policliniconapoli.it (S.G.); giuliorivetti94@gmail.com (G.R.); anna.disessa@unicampania.it (A.D.S.); deluciamaeva@gmail.com (M.D.L.); pieropalma2710@gmail.com (P.L.P.); emanuele.miraglia@unicampania.it (E.M.d.G.); cesare.polito@golfonet.it (C.P.)

**Keywords:** pediatric acute kidney injury, type 1 diabetes mellitus, creatinine, diagnostic performance

## Abstract

At type 1 diabetes mellitus (T1DM) onset, acute kidney injury (AKI) is very common. To diagnose AKI, the availability of a baseline serum creatinine (bSCr) is pivotal. However, in most hospitalized children the bSCr is unknown. We aimed to test whether the bSCr estimated on the basis of height (ebSCr) could be a reliable surrogate for AKI diagnosis compared with the measured bSCr (mbSCr). As the mbSCr, we considered the creatinine measured 14 days after T1DM onset while ebSCr (mg/dL) = (k × height [cm])/120 mL/min/1.73 m^2^, where k = 0.55 for children and adolescent girls and k = 0.7 for adolescent boys. AKI was defined as serum creatinine values >1.5 times the baseline creatinine. Kappa statistics and the percentage of agreement in AKI classification by ebSCr–AKI versus mbSCr–AKI definition methods were calculated. Bland–Altman plots were used to show the agreement between the creatinine ratio (highest/baseline creatinine; HC/BC) calculated with mbSCr and ebSCr. The number of 163 patients with T1DM onset were included. On the basis of mbSCr, 66/163 (40.5%) presented AKI while, on the basis of ebSCr, 50/163 (30.7%) accomplished AKI definition. ebSCr showed good correlation with mbSCr using both the Spearman test (rho = 0.67; *p* < 0.001) and regression analysis (r = 0.68; *p* < 0.001). Moreover, at the Bland–Altman plots, the bias of the highest/baseline creatinine ratio calculated on the basis of the mbSCr compared to ebSCr was minimal (bias = −0.08 mg/dL; 95% limits of agreement = −0.23/0.39). AKI determined using ebSCr showed 90% agreement with AKI determined using mbSCr (kappa = 0.66; *p* < 0.001). Finally, we compared the area under a receiver–operating characteristic curve (AUROC) of HC/BC ratio calculated on the basis of ebSCr with AUROC of the gold standard HC/BC ratio calculated on the basis of mbSCr. As expected, the gold standard had an AUROC = 1.00 with a 95% confidence interval (CI) between 0.98 and 1.00, *p* < 0.001. The HC/BC ratio calculated on the basis of ebSCr also had significant AUROC (AUROC = 0.94; 95% CI: 0.90–0.97; *p* < 0.001). The comparison of the two ROC curves showed a *p* < 0.001. In conclusion, when mbSCr is unknown in patients with T1DM onset, the ebSCr calculated on the basis of height could be an alternative to orientate clinicians toward AKI diagnosis.

## 1. Introduction

Acute kidney injury (AKI) is a common complication in several pediatric diseases, such as acute gastroenteritis or community-acquired pneumonia [1,2,3]. Unfortunately AKI is often underrecognized in children [4] and a nephrology follow-up after AKI in pediatric patients is still uncommon [5,6]. The AKI diagnosis, however, is important because even a mild AKI episode doubles the risk of chronic kidney disease (CKD) [7] and a specific follow-up has been associated with improved outcomes after severe kidney injury in adults [8].

The pediatric AKI under-recognition could be probably related to the lack of a known baseline serum creatinine value (bSCr) in most hospitalized children. The bSCr—according to the Kidney Disease/Improving Global Outcomes (KDIGO) guidelines regarding AKI—is important to stage the AKI. No AKI, in fact, is defined as occurring if serum creatinine values are <1.5 times the bSCr, stage 1 AKI if a creatinine value is 1.5 to <2, stage 2 if 2 to <3 and stage 3 if ≥3 times the bSCr [9]. To overcome the lack of a bSCr in most of pediatric cases, many methods of estimating bSCr have been developed in children [10]. The most common method to estimate bSCr is by back-calculation, using an estimated glomerular filtration rate (eGFR) equation and assuming normal baseline eGFR, traditionally eGFR = 120 mL/min/1.73 m^2^ [10].

In children with type 1 diabetes mellitus onset (T1DM), AKI is very common with an overall prevalence of about 45%, which could rise up to 65% in the case of diabetic ketoacidosis (DKA) [11,12].

According to the KDIGO criteria, AKI can also be defined by a reduction of urine output [9]. Since diabetic patients usually present polyuria and polydipsia during the onset of the disease, the urine output criterium seems to be less reliable for the diagnosis of AKI in this group of patients. In fact, only 15% of patients with AKI at T1DM onset met the urinary output KDIGO criteria [12].

Despite the high prevalence of AKI in children with T1DM onset, the diagnostic performance of the estimated bSCr toward the AKI diagnosis has never been investigated in this group of patients. Moreover, previous studies have used methods to estimate bSCr in children with T1DM without previous validation in this setting [11,13,14,15].

In a population of children admitted to pediatric intensive care units (PICU), Hessey et al. [10] demonstrated that the agreement when diagnosing AKI using both measured bSCr and bSCr estimated on the basis of patient’s height [16] was good. In this study, however, an heterogeneous group of patients admitted for any reason was enrolled [10]. Evidence indicates that T1DM diabetes represent only 2.8% of the PICU admissions in children [17], while reaching the 16.3% among non-PICU admissions [17]. Therefore, the data presented by Hessey et al. [10] appear insufficiently representative of T1DM children. In fact, the same authors stated that their data need of validation in different patients’ cohorts.

Taking advantage of the DiAKIdney study (onset of T1DM and AKI) prospective data collection and of the availability of the measured bSCr in all patients of this cohort [12], we aimed to test whether the bSCr estimated on the basis of height could be a reliable surrogate for AKI diagnosis compared with the measured bSCr in children with T1DM onset. 

Considering the high rate of AKI in patients with T1DM onset, the often lacking availability of a known bSCr in children and the lack of previous validation in children with T1DM onset, in our opinion, it is pivotal to test the diagnostic performance of the estimated bSCr (ebSCr) compared with the measured bSCr (mbSCr) in this setting in order to correctly diagnose an AKI episode in a category of patients at risk of CKD.

Correctly identifying an AKI episode in these patients, could, in fact, help in the selection of patients needing both a specific rehydration approach to avoid the progression of the kidney damage in the short term and a specific follow-up to improve the long-term outcomes. Moreover, this validation could be also useful for research purposes, considering that the literature about AKI in children with T1DM is constantly growing.

## 2. Methods

The DiAKIdney cohort is a prospectively enrolled a cohort of 185 children with T1DM onset, in which patients were evaluated both at T1DM onset and 14 days later when they had fully recovered from the acute phase [12]. The patients were admitted in a non-PICU ward with particular expertise in the management of T1DM patients.

Inclusion criteria of the DiAKIdney cohort were (i) onset of T1DM; (ii) age < 18 years; (iii) not on any medication apart from intravenous 0.9% NaCl infusion. Exclusion criteria were (i) not returning for the scheduled follow-up; (ii) anomalies of the kidney and urinary tract.

For the analyses of this manuscript we also excluded 22 patients aged <2 years in order to include only patients with a mature renal function [18]. Therefore, 163 patients were included in the analyses. None of these patients was admitted to PICU before or after admission to our ward.

All the patients had T1DM autoimmune diabetes with positive glutamic acid decarboxylase and/or islet antigen 2 and/or insulin and/or Zinc transporter 8 antibodies at the time of diagnosis.

The creatinine serum levels were determined at admission, every 24 h for first 3 days and then after 5 days and 14 days from the time of hospitalization.

### 2.1. AKI Definition

AKI was defined as serum creatinine values >1.5 times the baseline creatinine, according to KDIGO serum creatinine criteria [9]. We considered mbSCr as the creatinine measured at a follow-up visit 14 days after T1DM onset when patients had fully recovered from the acute phase [12]. This is because, in the DiAKIdney cohort, all the patients reverted AKI within 5 days with the exception of one patient who normalized kidney function after 14 days [12]. We also calculated the ebSCr using an height-based previously-validated back-calculation method [10]. In more detail: in our laboratories the creatinine has been dosed with Jaffe method (methodology: Alkaline Picrate, Abbott catalog no. 7D64-20) by using the Architect c16000 automated analyzer (Abbott Diagnostics Inc, Park City, IL, USA). For this reason, as previously adopted in other our studies [12,19,20], we used the old [21] instead of the most updated [16] Schwartz formula, considering that the original Schwartz equation was created using creatinine levels dosed using the Jaffe method. Therefore, the ebSCr was calculated as follows: ebSCr (mg/dL) = (k × height [cm])/eGFR. The k was 0.55 for children and adolescent girls, and 0.7 for adolescent boys [21]. Since we enrolled only children >2 years of age we assumed as a baseline eGFR an eGFR = 120 mL/min/1.73 m^2^ [10].

According to the KDIGO definition, we evaluated the presence of AKI on the basis of the ratio between the highest creatinine and the baseline creatinine (HC/BC ratio). To compare the diagnostic performance toward AKI diagnosis of the ebSCr, we compared the HC/BC ratio calculated using the mbSCr as denominator with the HC/BC ratio calculated using the ebSCr as denominator.

### 2.2. Post-Hoc Power Calculation

On the basis of the previously identified prevalence of 43.8% of AKI defined on the basis of mbSCr in children with T1DM onset [12], considering the prevalence of AKI defined on the basis of ebSCr of 30.7% in this population of 163 subjects, the calculated post hoc power, with an alpha of 0.05, was 93.8%.

### 2.3. Statistical Analysis

*p* values < 0.05 were considered significant. Data are shown as means ± standard deviation score (SDS). However, differences for continuous variables were analyzed using the independent sample *t* test for normally distributed variables and with the Mann–Whitney test in case of non-normality.

After log-transformation, because they were non-normally distributed variables, a linear regression analysis was performed to assess the relationship between mbSCr and ebSCr. The correlation between mbSCr and ebSCr by the Spearman test was assessed. Kappa statistics and the percentage of agreement in AKI classification (yes/no) by ebSCr–AKI versus mbSCr–AKI definition methods were calculated. Kappa statistics provided the Cohen’s kappa coefficient, which is a more robust measure than simple percent agreement calculation, as kappa takes into account the possibility of the agreement occurring by chance. There is controversy surrounding Cohen’s kappa due to the difficulty in interpreting indices of agreement. However, Landis and Koch characterized values < 0 as indicating no agreement and 0–0.20 as slight, 0.21–0.40 as fair, 0.41–0.60 as moderate, 0.61–0.80 as substantial, and 0.81–1 as almost perfect agreement [22]. Bland–Altman plots were used to show the agreement between the HC/BC ratio calculated by using mbSCr or ebSCr as a denominator. The Bland–Altman plot is used to quantify the bias and a range of agreement in measurements between two different instruments, within which 95% of the differences between one measurement and the other are included. If the line of equality (bias) is not in the interval, there is a significant systematic difference, i.e., the second method constantly under- or over-estimates compared to the first one. However, smaller the range of the agreement and the bias, the higher the concordance between the two techniques [23]. Moreover, the discriminating power of the ebSCr toward AKI was evaluated by comparing area under receiver–operating characteristic (AUROC) curve of HC/BC ratio, calculated on the basis of ebSCr with AUROC using the gold standard HC/BC ratio calculated on the basis of mbSCr. To compare the two AUROCs, the method described by Hanley and McNeil was used [24].

SPSS 25 software and GraphPad Prism 8.0 for Windows were used for all the statistical analyses.

## 3. Results

One hundred and sixty-three (seventy-two of male gender) patients with T1DM onset were included in the present study. The mean age was 9.9 years (3.4 SDS).

### 3.1. AKI Definition on the Basis of mbSCr

Out of 163 patients, 66 (40.5%) presented AKI. Separately evaluating patients with DKA, AKI was present in 51 out of 83 (61.4%) patients. Among patients without DKA at T1DM onset, 15 out of 80 (18.7%) had AKI.

### 3.2. AKI Definition on the Basis of ebSCr

Fifty out of 163 (30.7%) patients accomplished the AKI definition. Separately evaluating patients with DKA, AKI was detected in 40 out of 83 (48.2%) patients. Among patients without DKA at T1DM onset, 10 out of 80 (12.5%) had AKI.

### 3.3. Performance of ebSCr Compared with mbSCr in Diagnosing AKI

Comparing the HC/BC ratio calculated on the basis of the mbSCr and on the basis of ebSCr, the values and means were similar when comparing the two methods (means: 1.43 ± 0.38 SDS for mbSCr vs. 1.35 ± 0.36 SDS for ebSCr; *p* > 0.05) (Figure 1).

ebSCr showed good correlation with mbSCr both in the Spearman test (rho= 0.67; *p* < 0.001) and regression analysis (Figure 2).

The Bland–Altman graph (Figure 3) shows the values of both the superior and inferior limits and the average of the bias of the HC/BC ratio calculated on the basis of the mbSCr and of ebSCr. The bias was minimal (bias = −0.08 mg/dL; 95% limits of agreement from −0.23 to 0.39) (Figure 3). AKI determined using ebSCr showed 90% agreement with AKI using mbSCr (kappa = 0.66; *p* < 0.001).

Finally, we compared the AUROC of HC/BC ratio calculated on the basis of ebSCr with AUROC of the HC/BC ratio calculated on the basis of mbSCr (gold standard). As expected, the gold standard (HC/BC ratio calculated on the basis of mbSCr) had an AUROC = 1.00 with a 95% confidence interval (CI) between 0.98 and 1.00, *p* < 0.001 (Figure 4, blue line). The HC/BC ratio calculated on the basis of ebSCr also had significant AUROC (AUROC = 0.94; 95% CI: 0.90–0.97; *p* < 0.001) (Figure 4, dotted line). The comparison of the two ROC curves showed a *p* < 0.001.

## 4. Discussion

Patients with T1DM onset represent a category of children particularly at risk of developing kidney complications. In fact, during T1DM onset, these patients may experience AKI with subsequent increased risk of later CKD [7,11,12] and, during the following course of T1DM, they can develop diabetic nephropathy, especially in case of poor glycemic control [25].

The identification of T1DM patients who have developed an AKI episode is relevant for the planning of an accurate follow-up for renal function, microalbuminuria and blood pressure. For example, patients who developed an AKI at T1DM onset presented an increased risk of developing microalbuminuria during the follow-up [26].

The acute tubular necrosis represents the maximal expression of the kidney damage following kidney hypoperfusion at T1DM onset, which implies the shift of the AKI from functional to intrinsic [27,28]. For this reason, it is important to not miss the AKI diagnosis at T1DM onset or, in case of recurrent DKA, to identify patients that could benefit from a more “aggressive” rehydration schedule to promptly revert the AKI and to prevent the progression toward acute tubular necrosis, whilst paying attention to potential neurological complications. Laskin et al. suggested giving fluids to T1DM patients with AKI secondary to volume depletion, while quickly shifting to more restrictive strategies in those who do not respond to volume and have decreasing urinary output [29]. The data deriving from the study by Kuppermann et al., however, are reassuring concerning the risk of neurologic outcomes in patients with DKA [30]. In fact, neither the rate of administration nor the sodium content of intravenous fluids significantly influenced neurological outcomes in children with DKA [30].

The present study adds knowledge which could improve the care of patients with T1DM onset. Moreover, this study could be useful also for research purposes because the interest toward AKI in T1DM patients is constantly growing and previous studies used—without validation—ebSCr to evaluate the AKI prevalence [11,13,14,15]. Analyzing the diagnostic performance of the ebSCr (when used as denominator in the HC/BC ratio) in regard to AKI diagnosis in children with T1DM onset, we found that the ebSCr showed a 90% agreement with the gold standard mbSCr. Moreover, at ROC curve analysis, despite the AUROC of HC/BC ratio being calculated on the basis of mbSCr, was higher compared with the AUROC of the HC/BC ratio calculated on the basis of ebSCr (*p* < 0.001), the AUROC of the HC/BC ratio calculated on the basis of ebSCr was still reached the high value of 0.94, with 95% CI = 0.90–0.97 and *p* < 0.001.

This could indicate that when a mbSCr is lacking, such as in critically ill children in the case of PICU [10], it is possible to use an ebSCr on the basis of height to screen the patient with T1DM onset for possible AKI. 

However, we want to underline that to precisely back-calculate the creatinine on the basis of height, clinicians must know what method their laboratories use to measure creatinine. In fact, when using eGFR equations, it is imperative to remember that an equation performs best when applied to a similar patient population and using measurement methods that are equivalent to those used in the equation’s development. Therefore, when back-calculating creatinine on the basis of height, if the creatinine is measured with the IDMS-traceable method, the updated Schwartz equation should be preferred [ebSCr (mg/dL) = (0.413 × height [cm])/baseline eGFR]. On the other hand, if the creatinine is measured using the Jaffe method, the original Schwartz equation [21] should be adopted, as we did in the present manuscript. Moreover, as shown in the study of Hessey et al. [10] and in the present manuscript, the baseline eGFR that could be utilized in the formula is 120 mL/min/1.73 m^2^.

For the centers using the IDMS method to measure creatinine, further validation is needed by using the new Schwartz equation [16]. Meanwhile, both the findings of Hessey et al. [10], who used the IDMS method to measure creatinine and the new Schwartz equation [16] to back-calculate ebSCr in a population of children attending PICU for any reason, and of our study, who used the Jaffe method to measure creatinine and the old Schwartz equation [21] to back-calculate ebSCr in a population of children with T1DM onset, are reassuring in regard to a good diagnostic performance of both methods to back-calculate ebSCr in children with T1DM.

A strength of this study is the availability of measured bSCr of all enrolled children, allowing the comparison of ebSCr with the gold standard-measured bSCr in AKI diagnosis.

This study, however, has limitations. Firstly, it lacks an adequate number of patients aged <2 years and, therefore, it has not been possible to accurately test ebSCr for AKI diagnosis at T1DM onset in this age range. Secondly, the measurement of the creatinine using the Jaffe method did not allow us to test for AKI diagnosis using the baseline creatinine estimated on the basis of age [31]. This equation, in fact, was developed on the basis of creatinine measured with the IDMS-traceable method. Thirdly, the original Schwartz equation to calculate eGFR (such as the newer methods of eGFR calculation) has never been compared with the measured glomerular filtration rate by more accurate methods in children with T1DM onset. Fourthly, in the present study, as in other previous studies about this topic [11,12,13,15], we used serum creatinine as an indicator of AKI in patients with DKA, despite the fact that acetoacetates, hyperglycemia and glycosylated hemoglobin could falsely elevate the measured creatinine [14,32], and despite the fact that creatinine is a tardive marker of AKI [33]. Serum creatinine, however, is the only marker of renal function adopted in daily clinical practice, also in patients with DKA. Moreover, another factor that should be taken into consideration regarding the diagnostic accuracy of AKI in children with T1DM onset is that patients with T1DM can have a supraphysiologic elevation of eGFR due to glomerular hyperfiltration in the early stages [34]. Finally, the formulas to calculate eGFR can only be appropriately used when the renal function is in a stable steady state [21] and have not been designed to calculate eGFR in an acute kidney disease, such as AKI [35]. These formulas, however, have already been used in this condition in other previous studies [11,12,13,15].

In conclusion, we found that when mbSCr is unknown in patients with T1DM onset, the ebSCr calculated on the basis of height could be an alternative to orientate clinicians toward AKI diagnosis, while not missing those patients in need of a more aggressive rehydration approach and a specific post-discharge nephrology follow-up. This study is particularly useful for the centers performing creatinine measurement using the Jaffe method (still used in 2019 in about 24% of laboratories surveyed by the College of American Pathologists [36]), while the confirmation of this data regarding creatinine measurement using the IDMS method is needed.

## Figures and Tables

**Figure 1 children-09-00899-f001:**
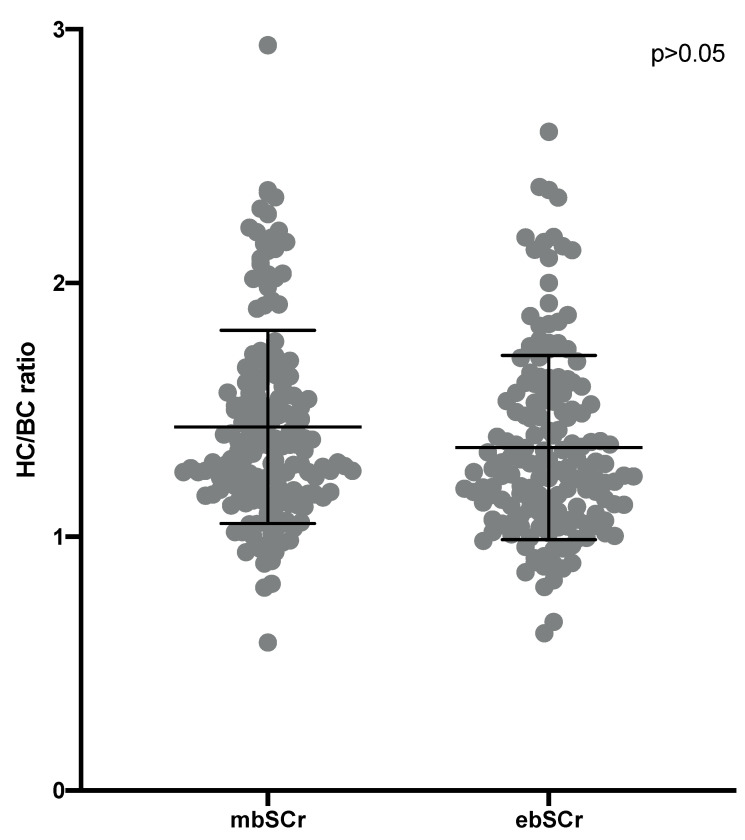
Highest/baseline creatinine (HC/BC) ratio calculated on the basis of the measured baseline creatinine (mbSCr) compared with the HC/BC ratio calculated on the basis of height-estimated baseline creatinine (ebSCr). Figure 1 legend: Means ± SDS are shown. However, these variables being non-normally distributed, they were compared by the Mann–Whitney test.

**Figure 2 children-09-00899-f002:**
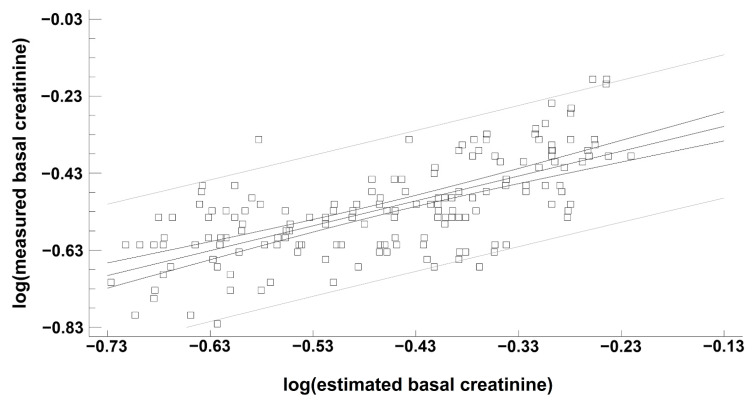
Regression analysis describing the relationship between measured baseline creatinine (mbSCr) and height-estimated baseline creatinine (ebSCr). Model r^2^ = 46.6 percent; *p* < 0.001; correlation coefficient = 0.68. The regression is described by the equation y = −0.224701 + 0.645172x. *p* value for intercepts was <0.001, *p* value for the slopes was <0.001. Spearman test: rho = 0.67, *p* < 0.001.

**Figure 3 children-09-00899-f003:**
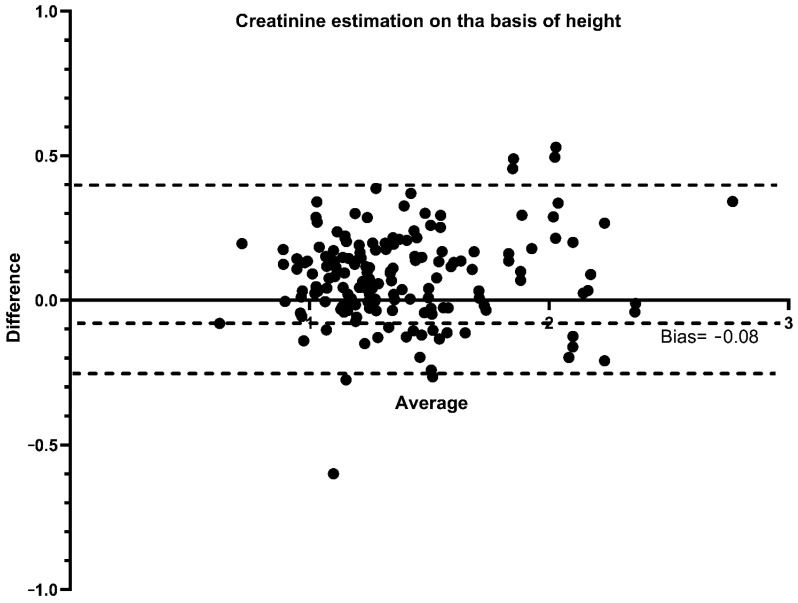
Bland–Altman plots comparing the HC/BC calculated on the basis of the measured baseline creatinine and on the basis of the height-estimated baseline creatinine. Dashed lines represent the limits of agreement and mean difference (bias) in estimations.

**Figure 4 children-09-00899-f004:**
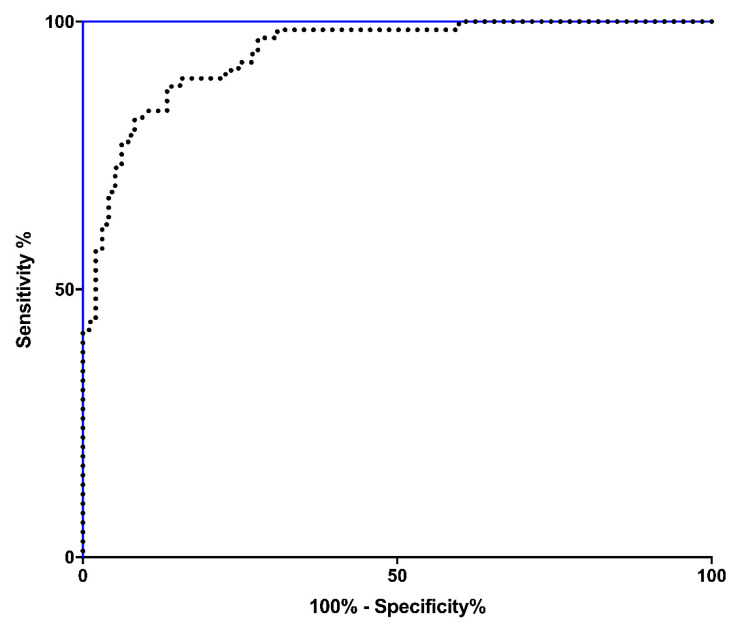
Comparison of the AUROC of the HC/BC ratio calculated on the basis of ebSCr with AUROC of the HC/BC ratio calculated on the basis of mbSCr (gold standard).

## Data Availability

Data supporting reported results can be obtained on request.

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
