# Peer review of "Diagnostic Performance of Height-Estimated Baseline Creatinine in Diagnosing Acute Kidney Injury in Children with Type 1 Diabetes Mellitus Onset"

_children, 2022, doi:10.3390/children9060899_

Round 1

Reviewer 1 Report

I think this report is well written as it examines whether the reference value of Cr (ebCr) by height when measuring sCr by the Jaffe method can be used in DM patients.

However, some questions remain.

Hessey et al. had developed a formula for calculating ebCr in pediatric patients admitted to the ICU, and you have validated whether this method can also be used in DM patients. You should emphasize this a little more. You cite the fact that there were very few DM patients at the time of the Hessey’s study as the basis for the present study. However, you should be more clear about your reason for thinking that the ebCr of DM patients may be different from that of other diseases.

In your comparison of ebCr and mbCr, you have done a paired t test and there is a significant difference, but this may be because ebCr is only an approximation. If the correlation between the two values is strong, there would be no need to consider the former study.

If you could fix this degree, I think the paper would be much better.

Author Response

We thank Editor and Reviewers for having spent their time in reviewing our manuscript and for having invited resubmission after minor revision.

Reviewer 1

I think this report is well written as it examines whether the reference value of Cr (ebCr) by height when measuring sCr by the Jaffe method can be used in DM patients.

Answer: thank you

However, some questions remain.

  1. Hessey et al. had developed a formula for calculating ebCr in pediatric patients admitted to the ICU, and you have validated whether this method can also be used in DM patients. You should emphasize this a little more. You cite the fact that there were very few DM patients at the time of the Hessey’s study as the basis for the present study. However, you should be more clear about your reason for thinking that the ebCr of DM patients may be different from that of other diseases.

Answer: we better specified why we decided to test the ebSCr in a selected population of children with T1DM onset in the new version of the manuscript. Please see lines 89-93 of the new version of the manuscript.

  1. In your comparison of ebCr and mbCr, you have done a paired t test and there is a significant difference, but this may be because ebCr is only an approximation. If the correlation between the two values is strong, there would be no need to consider the former study.

Answer: We are agreed with you about the no need to consider the paired t test because the correlation between ebSCr and mbSCr is strong. We deleted the information about the paired t test from the manuscript. Please see lines 29-31, 153-154, 199-202, and 276-278 of the new version of the manuscript. Moreover, following comment n° 3 of the Reviewer 2 we added new data about ROC curve analysis (please see lines 35-40, 171-175, 219-228, 281-285, and Figure 4 of the new version of the manuscript).

If you could fix this degree, I think the paper would be much better.

Answer: thank you for your comments.

Reviewer 2 Report

This is an interesting study that intends to evaluate the diagnostic performance of the estimated baseline creatinine of children with newly diagnosed Type 1 Diabetes mellitus based on their height.  The article is overall of good quality, however sometimes difficult to understand in terms of the writing and the description of the implemented methodology.

Among the keywords I suggest using diagnostic performance instead of diagnostic errors. Also including the pediatric population is relevant, e. g. in form of children or pediatric AKI, or something else.

Here are some of my observations:

Methods:

-         - A more thorough description of the interpretation of the kappa statistics (e. g. including kappa value interpretation gradings) and Bland-Altmann-plots is necessary.

-         -  Authors used comparative tests to prove their hypothesis (comparing diagnostic accuracy of measured serum creatinine and the height-based estimated serum creatinine). As far as I know, ROC Curves are also an interesting tool for assessing diagnostic accuracy/performance of tests. Wouldn’t it be interesting in this analysis to compare the ROC Curves between the ratios of measured and estimated serum creatinine values?

Results:

-          What test was used to compare the results from figure one? Authors have stated that the results were not normally distributed and that logarithmic transformation was necessary. Please specify.

-          If I understand correctly, the mean differences were compared between measured and estimated serum creatinine. If it is correct, the differences are expressed as mean and SD. However, the SD of both differences is very big (twice as the mean) making this a highly imprecise result and influencing its interpretation. Please reevaluate this. 

Discussion:

-          The whole part of the discussion explaining AKI and tubular damage/necrosis is not relevant in this study and should be substantially shortened.

-          Overall, authors should focus on the importance of their study and the consequences that it has in the diagnostic accuracy of AKI in diabetic children. The part regarding the Jaffe measurements is also unnecessarily extensive. 

Further observations:

-         - There are other markers important for diagnosing AKI, including for example oliguria. Was it possible to measure urine output in these patients? Could authors make a statement in terms of the presence of oliguria in pediatric patients with new onset of type 1 diabetes?

-         - Authors mention factors that falsely increase serum creatinine implying a worse kidney function. However, it is well known that patients with either type 1 or type 2 diabetes tend to have in early stages a supraphysiologic elevation of eGFR due to glomerular hyperfiltration. This should also be mentioned in the discussion and taken in to consideration in the diagnostic accuracy of AKI in children with new onset of type 1 Diabetes mellitus.

Author Response

We thank Editor and Reviewers for having spent their time in reviewing our manuscript and for having invited resubmission after minor revision.

Reviewer 2

This is an interesting study that intends to evaluate the diagnostic performance of the estimated baseline creatinine of children with newly diagnosed Type 1 Diabetes mellitus based on their height.  The article is overall of good quality, however sometimes difficult to understand in terms of the writing and the description of the implemented methodology.

  1. Among the keywords I suggest using diagnostic performance instead of diagnostic errors. Also including the pediatric population is relevant, e. g. in form of children or pediatric AKI, or something else. 

Answer: accordingly, we used diagnostic performance instead of diagnostic errors and we added information about the population including “pediatric AKI” among key-words. Please see lines 43-44 of the new version of the manuscript).

Here are some of my observations: 

Methods:

  1. A more thorough description of the interpretation of the kappa statistics (e. g. including kappa value interpretation gradings) and Bland-Altmann-plots is necessary. 

Answer: We modified the text accordingly. Kappa statistics provides the Cohen's kappa coefficient which is a more robust measure than simple percent agreement calculation, as kappa takes into account the possibility of the agreement occurring by chance. There is controversy surrounding Cohen's kappa due to the difficulty in interpreting indices of agreement. However, Landis and Koch characterized values < 0 as indicating no agreement and 0–0.20 as slight, 0.21–0.40 as fair, 0.41–0.60 as moderate, 0.61–0.80 as substantial, and 0.81–1 as almost perfect agreement (Please see lines 156-162 of the new version of the manuscript).

The Bland-Altman plot is used to quantify the bias and a range of agreement in measurements between two different instruments, within which 95% of the differences between one measurement and the other are included. If the line of equality (bias) is not in the interval, there is a significant systematic difference, i.e. the second method constantly under- or over- estimates compared to the first one. However, smaller is the range of the agreement and the bias, higher is the concordance between the two techniques (Please see lines 164-170 of the new version of the manuscript).

  1. Authors used comparative tests to prove their hypothesis (comparing diagnostic accuracy of measured serum creatinine and the height-based estimated serum creatinine). As far as I know, ROC Curves are also an interesting tool for assessing diagnostic accuracy/performance of tests. Wouldn’t it be interesting in this analysis to compare the ROC Curves between the ratios of measured and estimated serum creatinine values?

 Answer: Thank you for your comment, in our opinion, the suggested analysis improved the data presentation. We added the required analysis. We presented the new analyses in the method, results and discussion section (please see lines 35-40, 171-175, 219-228, 281-285, and Figure 4 of the new version of the manuscript).

Results:

  1. What test was used to compare the results from figure one? Authors have stated that the results were not normally distributed and that logarithmic transformation was necessary. Please specify. If I understand correctly, the mean differences were compared between measured and estimated serum creatinine. If it is correct, the differences are expressed as mean and SD. However, the SD of both differences is very big (twice as the mean) making this a highly imprecise result and influencing its interpretation. Please reevaluate this. 

Answer: we clarified this aspect. The data are shown as means ± standard deviation score and because non-normally distributed they were compared by Mann-Whitney test. Please see lines 147-148 and 195-196 of the new version of the manuscript.

Discussion:

  1. The whole part of the discussion explaining AKI and tubular damage/necrosis is not relevant in this study and should be substantially shortened. 

Answer: we substantially shortened this section, please see lines 235-272 of the new version of the manuscript.

  1. Overall, authors should focus on the importance of their study and the consequences that it has in the diagnostic accuracy of AKI in diabetic children. The part regarding the Jaffe measurements is also unnecessarily extensive. 

Answer: we substantially shortened also this section. Please see lines 289-324 and 354-355 of the new version of the manuscript.

Further observations: 

  1. There are other markers important for diagnosing AKI, including for example oliguria. Was it possible to measure urine output in these patients? Could authors make a statement in terms of the presence of oliguria in pediatric patients with new onset of type 1 diabetes?

Answer: we added this information in the new version of the manuscript. Please see lines 67-71 of the new version of the manuscript.

  1. Authors mention factors that falsely increase serum creatinine implying a worse kidney function. However, it is well known that patients with either type 1 or type 2 diabetes tend to have in early stages a supraphysiologic elevation of eGFR due to glomerular hyperfiltration. This should also be mentioned in the discussion and taken in to consideration in the diagnostic accuracy of AKI in children with new onset of type 1 Diabetes mellitus.

Answer: we added this consideration in the new version of the manuscript. Please see lines 341-344 of the new version of the manuscript.

This manuscript is a resubmission of an earlier submission. The following is a list of the peer review reports and author responses from that submission.

Round 1

Reviewer 1 Report

Sorry for the dealuy in review, 

overall I think this is a good paper, it is incremental knowledge on AKI, but is useful to publish. The original papers on AKI are quite biased, and were in children with established diabetes with what we would not consider good control, so its reassuring that the est baseline creatinine data is actually relatively robust.

I am not a fan of too many abbrevations and got lost briefly with the low/high creatinine one, so wonder if that is useful .

It has a good number of subjects, is prospective, does not over state the implications and my only caveat is I would like a stats review, esp on the log aspects of the data etc (I am not a statistician).

Author Response

Reviewer 1

Overall I think this is a good paper, it is incremental knowledge on AKI, but is useful to publish. The original papers on AKI are quite biased, and were in children with established diabetes with what we would not consider good control, so its reassuring that the estimated baseline creatinine data is actually relatively robust.

Answer: thank you

I am not a fan of too many abbreviations and got lost briefly with the low/high creatinine one, so wonder if that is useful.

Answer: According to the KDIGO definition, we evaluated the presence of AKI on the basis of the ratio between the highest creatinine and the baseline creatinine (HC/BC ratio). To compare the diagnostic performance toward AKI diagnosis of ebSCr, we compared the HC/BC ratio calculated using as denominator the mbSCr with the HC/BC ratio calculated using as denominator the ebSCr. For this reason, we think that it is important to maintain the HC/BC abbreviation. We gave more information about the HC/BC ratio in the new version of the manuscript to avoid giving misleading information to the Readers and to better clarify this acronymous. Please see lines 109-113 of the new version of the manuscript.

It has a good number of subjects, is prospective, does not overstate the implications and my only caveat is I would like a stats review, esp on the log aspects of the data etc (I am not a statistician).

Answer: All our analyses were reviewed by our expert in statistics.

Reviewer 2 Report

This study investigates the performance of height estimated baseline creatinine in the diagnosis of AKI in children with type 1 diabetes mellitus. I have the following comments:

  1. I would recommend to replace ‘basal’ by ‘baseline’. Basal rate is the rate of continuous supply of some chemical or process. In diabetes mellitus, it is a low rate of continuous insulin supply needed for such purposes of controlling cellular glucose and amino acid uptake. I don’t think this is what is meant here. To define AKI, a baseline serum creatinine is required and is often not available.
  2. It seems strange to me to calculate a baseline SCr from SCr = k x Ht / 120, which is based on the Schwartz equation, which has been developed for CKD children with growth retardation. Apparently, the authors still use the outdated Schwartz equation, developed for SCr measured by Jaffe assays, as they use k = 0.55 for children and k = 0.7 for adolescent boys. Most modern SCr assays, also Jaffe type assays, are now traceable to the IDMS gold standard, and this also means that the old Schwartz equation should not be used.
  3. Back calculation from eGFR-formulas is not a good idea. First of all, you need to ‘assume’ the average GFR of a child to be able to do the back-calculation. E.g. you may choose 120 mL/min/1.73m², but this may be wrong, probably too high. The pediatric form of the FAS-equation is eGFR = 107.3 / [SCr/Q] which means that the average GFR is assumed to be 107.3 mL/min/1.73m², thus, about 12% lower than the value assumed by the authors. Second, such a back calculation is also based on a specific formula, which can be wrong (see e.g. Pottel H. et al Evaluation of the creatinine-based chronic kidney disease in children (under 25 years) equation in healthy children and adolescents. Pediatr Nephrol. 2022 Jan 24. doi: 10.1007/s00467-022-05429-0. ), in which Pottel et al claim that the Schwartz equation has some built-in flaws, when applied to healthy children, like the systematic decline in eGFR with age. Why not using a direct relationship between SCr and height for the baseline SCr?
  4. Baseline SCr can be obtained directly from studies that have described the relationship between SCr and age, or SCr and height. See e.g. Hoste et al. A new equation to estimate the glomerular filtration rate in children, adolescents and young adults. Nephrol Dial Transplant 2013; 29: 944–947. Hoste used median SCr from a large hospital database and linked it to the national growth curves to establish a relationship between median SCr and height of healthy children.
  5. Minor: Jaffe is without accent!!
  6. I don’t think you can compare the older generation of the Jaffe methods to the new generation of Jaffe methods, as all manufacturers try to calibrate their assays to the IDMS gold standard method, which was not the case for the old generation of Jaffe assays. The authors should look up in the insert, or contact the manufacturer, to know whether or not their Jaffe assay was calibrated to the IDMS gold standard method.
  7. The problem with the current study is that there is no way to evaluate if ebSCr can be used as the baseline value, as the authors assume that the renal function has fully recovered 14 days after the acute phase. Is this really so? Is there evidence for this?
  8. When comparing ebSCr with mbSCR the authors should present the mean and stdev of the paired differences, not for the variables separately, as these variables are not independent. Comparison should be based on a paired t-test. Figure 1 is misleading. Unless I do not understand, but the highest SCr is the same value when calculating the ratios for mbSCr and ebSCr. So why not simply presenting ebSCr and mbSCr, or even better, the authors should present the box plot of paired differences.
  9. What is the value of the regression equation? What can we learn from this? The slope should be ‘1’ and the intercept should be ‘0’. This is absolutely not the case.
  10. Why is mbSCr called HC? Why is that the highest creatinine?
  11. The Bland-Altman plot shows a systematic bias of -0.08 mg/dL, but again why HC/BC and not simply ebSCr – mbSCr versus the average?

Author Response

Reviewer 2

This study investigates the performance of height estimated baseline creatinine in the diagnosis of AKI in children with type 1 diabetes mellitus. I have the following comments:

  1. I would recommend to replace ‘basal’ by ‘baseline’. Basal rate is the rate of continuous supply of some chemical or process. In diabetes mellitus, it is a low rate of continuous insulin supply needed for such purposes of controlling cellular glucose and amino acid uptake. I don’t think this is what is meant here. To define AKI, a baseline serum creatinine is required and is often not available.

Answer: we modified the text accordingly.

  1. It seems strange to me to calculate a baseline SCr from SCr = k x Ht / 120, which is based on the Schwartz equation, which has been developed for CKD children with growth retardation. Apparently, the authors still use the outdated Schwartz equation, developed for SCr measured by Jaffe assays, as they use k = 0.55 for children and k = 0.7 for adolescent boys. Most modern SCr assays, also Jaffe type assays, are now traceable to the IDMS gold standard, and this also means that the old Schwartz equation should not be used.

Answer: thank you for your comment. This underlines a crucial issue of the study and it is important to better clarify it for a complete understanding. When using eGFR equations, it is imperative to remember that an equation performs best when using measurement methods that are equivalent to those used in the equation’s development. The new Schwartz equations were developed using data from children with chronic kidney disease and an isotope dilution mass spectrometry (IDMS)-traceable enzymatic creatinine method. The revised bedside Schwartz equation [(0.413 × Height (cm)) / Scr (mg/dL)] is recommended by the National Kidney Disease Education Program for use when creatinine is dosed IDMS methods. Changes in creatinine measurement and calibration over time caused the original Schwartz equation to overestimate GFR. Use of an IDMS-traceable creatinine value with the original Schwartz equation will overestimate GFR by 20–40% (https://www.kidney.org/content/creatinine-based-“bedside-schwartz”-equation-2009). This has also been confirmed by Srivastava et al. (Pediatr Res 65: 113–116, 2009) showing that there was a significant bias in IDMS-traceable methods when compared with Jaffè method underlining the need of new formula to calculate eGFR to avoid this bias.

However, our laboratories, at the present time, use the Jaffe method yet. We gave more information about the creatinine measurement in the new version of the manuscript, please see lines 100-102 of the new version of the manuscript. Therefore, the Jaffe method has been used to evaluate the serum creatinine and then we used the original Schwartz formula to calculate the eGFR. This because the original Schwartz equation has been created using creatinine levels dosed with the Jaffe method. Therefore, conscious of the intrinsic limits deriving from the use of a formula to estimate GFR, we think that in our cohort the best method to calculate eGFR is the old Schwartz equation because all the creatinine levels have been dosed with the Jaffe method (please see lines 256-272 and 281-283 of the new version of the manuscript).

  1. Back calculation from eGFR-formulas is not a good idea. First of all, you need to ‘assume’ the average GFR of a child to be able to do the back-calculation. E.g. you may choose 120 mL/min/1.73m², but this may be wrong, probably too high. The pediatric form of the FAS-equation is eGFR = 107.3 / [SCr/Q] which means that the average GFR is assumed to be 107.3 mL/min/1.73m², thus, about 12% lower than the value assumed by the authors. Second, such a back calculation is also based on a specific formula, which can be wrong (see e.g. Pottel H. et al Evaluation of the creatinine-based chronic kidney disease in children (under 25 years) equation in healthy children and adolescents. Pediatr Nephrol. 2022 Jan 24. doi: 10.1007/s00467-022-05429-0. ), in which Pottel et al claim that the Schwartz equation has some built-in flaws, when applied to healthy children, like the systematic decline in eGFR with age. Why not using a direct relationship between SCr and height for the baseline SCr?

Answer: Thank you for having raised another pivotal issue which needs to be clarified. The Pottel FAS-equation has been designed on the basis of creatinine measurement by IDMS method differently from our study in which creatinine was measured by Jaffe method (Alkaline Picrate method) (please see lines 100-102 of the new version of the manuscript). For this reason, the use of this equation is not applicable in our manuscript. In the paper from Hessey et al. (Pediatr. Nephrol. 2017, 32, 1953–1962, doi:10.1007/s00467-017-3670-z), Height-dependent (revised bedside Schwartz equation) and height-independent (FAS equation) bSCr estimation methods were comparable. To perform this study, Hessey et al., to back calculate the estimated basal creatinine assumed a baseline eGFR=120 mL/min/1.73m2 for children >2 years and for this reason and because we enrolled only patients >2years of age, we assumed as baseline eGFR and eGFR of 120 mL/min/1.73m2. Moreover, also in a previous paper published on JAMA Pediatrics, an estimated baseline eGFR value of 120 mL/min/1.73m2 was used to back calculated estimated bSCr in children with diabetic ketoacidosis (JAMA Pediatr. 2017;171(5):e170020. doi:10.1001/jamapediatrics.2017.0020).

To be consistent with these papers we decided to maintain the same method of Hessey et al. (estimated baseline eGFR=120 mL/min/1.73m2) to back-calculate bSCr.

While the updated Schwartz equation (CKiD equation) was designed for children with chronic kidney disease, the old Schwartz equation can be used to estimate GFR in infants, children, and adolescents without chronic kidney disease (Pediatr. Clin. North Am. 1987, 34, 571–90).

However, following your comment we specified among limitations of the study that the original Schwartz equation to calculate eGFR (such as the newer methods of eGFR calculation) has been never compared with the measured glomerular filtration rate by more accurate methods in children with T1DM onset and that the formulas to calculate eGFR can be appropriately used only when the renal function is in stable steady state and have not been designed to calculate eGFR in an acute kidney disease such as AKI. Please see lines 274-292 of the new version of the manuscript.

  1. Baseline SCr can be obtained directly from studies that have described the relationship between SCr and age, or SCr and height. See e.g. Hoste et al. A new equation to estimate the glomerular filtration rate in children, adolescents and young adults. Nephrol Dial Transplant 2013; 29: 944–947. Hoste used median SCr from a large hospital database and linked it to the national growth curves to establish a relationship between median SCr and height of healthy children.

Answer: Hessey et al. (Pediatr. Nephrol. 2017, 32, 1953–1962, doi:10.1007/s00467-017-3670-z) to compare height-dependent (revised bedside Schwartz equation) and height-independent (FAS equation) bSCr estimation methods to back calculate the estimated basal creatinine assumed a baseline eGFR=120 mL/min/1.73m2 for children >2 years and for this reason and because we enrolled only patients >2years of age, we assumed as baseline eGFR and eGFR of 120 mL/min/1.73m2. Moreover, also in a previous paper published on JAMA Pediatrics, an estimated baseline eGFR value of 120 mL/min/1.73m2 was used to back calculated estimated bSCr (JAMA Pediatr. 2017;171(5):e170020. doi:10.1001/jamapediatrics.2017.0020).

To be consistent with these papers we decided to maintain the same method of Hessey et al. (estimated baseline eGFR=120 mL/min/1.73m2) to back-calculate bSCr.

  1. Minor: Jaffe is without accent!!

Answer: we modified the text accordingly.

  1. I don’t think you can compare the older generation of the Jaffe methods to the new generation of Jaffe methods, as all manufacturers try to calibrate their assays to the IDMS gold standard method, which was not the case for the old generation of Jaffe assays. The authors should look up in the insert, or contact the manufacturer, to know whether or not their Jaffe assay was calibrated to the IDMS gold standard method.

Answer: we gave more information about the used Jaffe method in the new version of the manuscript. Please see lines 100-102 of the new version of the manuscript. This is not calibrated to IDMS. This is the reason why we decided to use the original Schwartz equation in this paper such as in our previous paper involving the eGFR evaluation (J. Clin. Endocrinol. Metab. 2021, 106, e2720–e2737, doi:10.1210/clinem/dgab090; J. Perinatol. 2019, 39, 129–134, doi:10.1038/s41372-018-0260-2; J. Ren. Nutr. 2018, 28, 359–362, doi:10.1053/j.jrn.2018.01.001).

  1. The problem with the current study is that there is no way to evaluate if ebSCr can be used as the baseline value, as the authors assume that the renal function has fully recovered 14 days after the acute phase. Is this really so? Is there evidence for this?

Answer: this study derives from the data collection of the DiAKIdney study (J. Clin. Endocrinol. Metab. 2021, 106, e2720–e2737, doi:10.1210/clinem/dgab090) a prospective study investigating the prevalence of AKI and of renal tubular damage in a cohort of patients with T1DM onset. All the patients recovered from AKI within 14 days and this has been accurately described in the above-mentioned manuscript. More in detail all the patients reverted AKI within 5 days with the exception of one patient who normalized kidney function after 14 days. We added this information in the text, please see lines 96-98 of the new version of the manuscript.

To give to you more information, we compared creatinine values at AKI diagnosis and after 14 days. At AKI diagnosis serum creatinine values were 0.87mg/dL±0.25SDS (range eGFR: 46.2-165.0mL/min/1.73m2) Vs 0.59mg/dL±0.07SDS (range eGFR: 100-185 mL/min/1.73m2) after 14 days; p<0.001.

  1. When comparing ebSCr with mbSCR the authors should present the mean and stdev of the paired differences, not for the variables separately, as these variables are not independent. Comparison should be based on a paired t-test. Figure 1 is misleading. Unless I do not understand, but the highest SCr is the same value when calculating the ratios for mbSCr and ebSCr. So why not simply presenting ebSCr and mbSCr, or even better, the authors should present the box plot of paired differences.

Answer: thank you for this important comment. We added the analyses deriving from paired t-test. We found that the null hypotheses of equal means of both mbSCr and ebSCr (paired differences = -0.03 ± 0.06; t = 7.91; p<0.001) and of HC/BC ratio calculated on the basis of mbSCr and ebSCr (paired differences = 0.08 ± 0.16; t = 6.5; p<0.001) were rejected.

Because AKI is defined by the ratio between the highest creatinine and the basal creatinine, we decided to not delete the Figure 1 to give an idea to the readers of the distribution of the HC/BC values calculated on the basis of mbSCr or ebSCr. However, we emphasize both in the abstract, results and discussion section that the null hypotheses were rejected at paired t-test. Please see lines 30-32, 150-153 and 247-251 of the new version of the manuscript. We also mitigated conclusions both in the abstract and in the manuscript (please see lines 37 and 295 of the new version of the manuscript.

  1. What is the value of the regression equation? What can we learn from this? The slope should be ‘1’ and the intercept should be ‘0’. This is absolutely not the case.

Answer: with this figure we want to show that a good correlation between mbSCr and ebSCr exists. In other words, increasing the mbSCr also the ebSCr increases. Also if this information is not clinically useful, we decided to not remove this figure to give also this statistical information to the Readers.

  1. Why is mbSCr called HC? Why is that the highest creatinine?

Answer: the mbSCr is not the HC. The HC is the highest creatinine during hospitalization. AKI, in fact, is defined by the ratio between the highest creatinine and the basal creatinine. We give more information about the HC/BC ratio in the new version of the manuscript to avoid giving misleading information to the Readers. Please see lines 109-113 of the new version of the manuscript.

  1. The Bland-Altman plot shows a systematic bias of -0.08 mg/dL, but again why HC/BC and not simply ebSCr – mbSCr versus the average?

Answer: because AKI is defined by the ratio between the highest creatinine and the basal creatinine we prefer to show this Bland-Altman plot to give to the Readers an idea of the ratios agreement using ebSCr or mbSCr as denominator.

However, following your comment we calculated the Bland-Altman plot for ebSCr  – mbSCr versus the average. The results were identical to those of the HC/BC calculated on the basis of the measured baseline creatinine – HC/BC calculated on the basis of height-estimated baseline creatinine. This because the HC was the same for both the ratios.

Reviewer 3 Report

This topic covers an important message: how to objectivate AKi in children with IDDM, when you have no basal values of creatinine, and the results have definitely an added value

  • introduction is well written
  • data are well presented
  • message is clear

But I miss several points in the discussion

  • creatinine is not stable, during an acute phase of AKI... thereby underestimating the AKI, and formula's of eGFR can only be appropriately used in stable steady state renal function . 
  • words like Acute tubular necrosis are used, but how frequent is this, have the authors follow up in those patients
  • AKI is  more than decreased GFR
  •  
    • since creatinine excretion is in prerenal AKI stimulated, thereby overestimating eGFR, but is this a also the case in IDDM,
    • Is there tubular dysfunction by the high glucose load
  • How to differentiate between renal an prerrenal
  • was there a correlation with the duration of clinical symptoms, and or degree of glucosuria ?

Author Response

Reviewer 3

This topic covers an important message: how to objectivate AKI in children with IDDM, when you have no basal values of creatinine, and the results have definitely an added value

  • introduction is well written
  • data are well presented
  • message is clear

Answer: Thank you.

But I miss several points in the discussion

  • creatinine is not stable, during an acute phase of AKI... thereby underestimating the AKI, and formula's of eGFR can only be appropriately used in stable steady state renal function . 

Answer: we added these considerations among limitations of the study. Please see lines 274-292 of the new version of the manuscript.

  • words like Acute tubular necrosis are used, but how frequent is this, have the authors follow up in those patients

Answer: we gave more information about AKI pathophysiology and acute tubular necrosis in the new version of the manuscript. Please see lines 214-234 of the new version of the manuscript.

  • AKI is  more than decreased GFR

Answer: following this comment and your previous comment we gave more information about AKI pathophysiology in the new version of the manuscript. Please see lines 214-234 of the new version of the manuscript.

  •  since creatinine excretion is in prerenal AKI stimulated, thereby overestimating eGFR, but is this a also the case in IDDM

Answer: following your comment we specified that formulas to calculate eGFR have not been designed for acute kidney diseases such as AKI (please see lines 289-292 of the new version of the manuscript). We are agreed with you that at low GFR levels tubular secretion of creatinine occurs and therefore CrCl may overestimate GFR (Clin Biochem Rev. 2016 Dec; 37(4): 153–175). However, we did not specify this concept in the new version of the manuscript to not confuse the Readers because our paper refer to eGFR and not to creatinine clearance using urinary collection.

  • Is there tubular dysfunction by the high glucose load

Answer: in a previous study we have demonstrated that a general tubular damage can be observed in up to 73.5% of patients with T1DM onset (doi:10.1210/clinem/dgab090). Of these, 32.4% may present with an acute tubular necrosis, while acute tubular damage and prerenal-AKI can be observed respectively in the 27.9% and 11.4% of the patients (doi:10.1210/clinem/dgab090). More the dehydration and acidosis are severe higher is the probability that a patient with T1DM onset develop an AKI and that, in turn, it progresses toward acute tubular necrosis (doi:10.1210/clinem/dgab090). Therefore, more than to high glucose load this tubular dysfunction is related to hypovolemia and acidosis. We added this information in the new version of the manuscript. Please see lines 214-234 of the new version of the manuscript.

  • How to differentiate between renal an prerenal

Answer: we gave more details about this issue in the new version of the manuscript. Please see lines 225-228 of the new version of the manuscript.

  • was there a correlation with the duration of clinical symptoms, and or degree of glucosuria ?

Answer: we do not have data to answer to this interesting question. In the DiAKIdney study we found that a longer duration of clinical symptoms was associated to higher risk of developing AKI indicating that a prompt T1DM diagnosis could prevent the development of the kidney complications (doi:10.1210/clinem/dgab090)

Round 2

Reviewer 2 Report

Our local hospital switched from the Jaffe type assay to the IDMS equivalent enzymatic SCr assay in 2003, nearly 20 years ago!! In the meantime, most clinical laboratories have made that transition and now use IDMS traceable methods to measure SCr. This is a basic requirement to make your research work useful for others. When your research is based on a method that is not IDMS traceable, then, what is the value of your research? Basically, only your own lab can make use of the results. 

Author Response

Reviewer 2

Our local hospital switched from the Jaffe type assay to the IDMS equivalent enzymatic SCr assay in 2003, nearly 20 years ago!! In the meantime, most clinical laboratories have made that transition and now use IDMS traceable methods to measure SCr. This is a basic requirement to make your research work useful for others. When your research is based on a method that is not IDMS traceable, then, what is the value of your research? Basically, only your own lab can make use of the results. 

Answer: as indicated by CDC not all the American laboratories were using IDMS-traceable calibration of serum creatinine (https://nccd.cdc.gov/ckd/detail.aspx?Qnum=Q223). In fact, as far as 24% of American laboratories were using the old method of creatinine measurement in 2019.

Moreover, Kume et al. in 2018 (J Clin Lab Anal. 2018;32:e22168) evaluated and compared the analytical performance characteristics of the two creatinine methods based on the Jaffe and enzymatic methods with the Architect c16000 automated analyzer (the same analyzer used in our laboratories). They found that both Jaffe and enzymatic methods were found to meet the analytical performance requirements in routine use. However, enzymatic method was found to have better performance in low creatinine levels. In the present study, because patients are dehydrated and the creatinine levels tend to be higher both methods of creatinine could be useful.

Moreover, in a previous our study enrolling children with congenital solitary kidney (J Ren Nutr. 2020 May;30(3):261-267. doi: 10.1053/j.jrn.2019.07.003) made in collaboration with another Italian center adopting the IDMS method to measure creatinine, to calculate the eGFR we used the original Schwartz formula if creatinine was estimated using the Jaffe method (in patients belonging to our centre), and the modified Schwartz formula if creatinine was measured using the isotope dilution mass spectrometry traceable method (in patients belonging to the other centre). Interestingly, creatinine levels of the 2 cohorts (patients of our center Vs patients of the other centre) were comparable: 0.55 mg/dL (0.17 SDS) for the IDMS method and 0.58 mg/dL (0.19 SDS) for Jaffe method; p=0.30. Moreover, also the eGFR levels of the 2 cohorts were comparable: 131.0 mL/min/1.73m2 (24.2SDS) for the new Schwartz equation and 131.8 mL/min/1.73m2 (25.7 SDS) for the old Schwartz equation, p=0.9. These findings are in line with the above-mentioned paper by Kume et al. and add evidence that using the right equation to the right method to measure creatinine possible differences disappear. This data has not previously shown and has been shown only here for your convenience. We have decided to not add this data in this manuscript because belonging to a different cohort of patients, and because ­–in case of the paper publication– Readers could find this information in the Answers to Reviewers because we have chosen open Review policy.

However, according to your comment, in the new version of the manuscript we have specified the following concept in the conclusions section (please see lines 291-298 of the new version of the manuscript):

 “This study is particularly useful for the centers performing creatinine measurement by Jaffe method while for the centers using IDMS method to measure creatinine further validation is needed. However, considering that in the study by Hessey et al. in children admitted to ICU [9] the performance of the back calculation of ebSCr using the new Schwartz equation [16] was good, the possibility exists that also in patients with T1DM onset –similarly to the data shown in the present manuscript– the obtainment of the ebSCr by back calculation with new Schwartz formula could be equally useful and effective to diagnose AKI.”

Reviewer 3 Report

All questions were well and complete answered an incorporated in the text. the response was nuanciated, and the paper improved significantly

Author Response

Reviewer 3

All questions were well and complete answered and incorporated in the text. the response was enunciated, and the paper improved significantly

Answer: thank you for your efforts in reviewing our manuscript.